# Glucose metabolism induced by Bmp signaling is essential for murine skeletal development

Seung-Yon Lee [1], E. Dale Abel [2] & Fanxin Long[1,3,4]

Much of the mammalian skeleton originates from a cartilage template eventually replaced by bone via endochondral ossification. Despite much knowledge about growth factors and nuclear proteins in skeletal development, little is understood about the role of metabolic regulation. Here we report that genetic deletion of the glucose transporter Glut1 (Slc2a1), either before or after the onset of chondrogenesis in the limb, severely impairs chondrocyte proliferation and hypertrophy, resulting in dramatic shortening of the limbs. The cartilage defects are reminiscent to those caused by deficiency in Bmp signaling. Importantly, deletion of Bmpr1a in chondrocytes markedly reduces Glut1 levels in vivo, whereas recombinant BMP2 increases Glut1 mRNA and protein levels, boosting glucose metabolism in primary chondrocytes. Biochemical studies identify a Bmp-mTORC1-Hif1a signaling cascade resulting in upregulation of Glut1 in chondrocytes. The results therefore uncover a hitherto unknown connection between Bmp signaling and glucose metabolism in the regulation of cartilage development.

[1] Department of Orthopaedic Surgery, Washington University School of Medicine, St. Louis, MO 63110, USA. [2] Fraternal Order of Eagles Diabetes Research Center and Division of Endocrinology and Metabolism, Carver College of Medicine, University of Iowa, Iowa City, USA. [3] Department of Developmental Biology, Washington University School of Medicine, St. Louis, MO 63110, USA. [4] Present address: The Children's Hospital of Philadelphia and University of Pennsylvania, Philadelphia, PA 19104, USA. Correspondence and requests for materials should be addressed to F.L. (email: longf1@email.chop.edu)

A majority of the bony skeleton in mammals, including long bones in the limbs, is derived from a cartilage template through endochondral ossification. During embryonic limb development, mesenchymal cells descended from the lateral plate mesoderm undergo chondrogenesis to form a cartilage anlage composed of chondrocytes surrounded by several layers of fibroblastic cells in the perichondrium[1,2]. The chondrocytes initially all undergo proliferation, but subsequently those residing at center of the cartilage anlage exit the cell cycle and undergo cellular hypertrophy. Recent studies have indicated that chondrocyte hypertrophy encompasses three distinct phases, including two with proportional increases in dry mass production and fluid uptake together with an intervening phase of cell swelling[3]. Although the terminally hypertrophic chondrocytes are traditionally believed to all undergo apoptosis, recent evidence indicates that a subpopulation may survive to produce osteoblasts[4–6]. The hypertrophic cartilage is subsequently invaded by blood vessels that not only initiate the formation of the bone marrow but also introduce osteoblast progenitors from the perichondrium for endochondral bone formation[7]. Once the bone marrow cavity is established, chondrocytes form the growth plate composed of discreet domains including proliferation and hypertrophy zones, at either end of the cartilage template. The proliferating chondrocytes can be further distinguished as round cells near the ends versus flat cells next to the hypertrophy region. As the flat cells are stacked into columns along the longitudinal axis, they are often referred to as columnar cells. While the cartilage template continues to grow at the ends due to proliferation and hypertrophy, it is progressively replaced by bone and the expanding marrow cavity from the middle, following resorption of the hypertrophic cartilage. Disruption of the normal sequence of cartilage development is well documented to cause a myriad of skeletal dysplasias in humans[8]. Thus, proper control and coordination of chondrocyte proliferation and hypertrophy is critical for ensuring the normal size and morphology of the skeleton.

Multiple growth factors have been implicated in regulating chondrocyte proliferation and hypertrophy during endochondral bone development[1]. For example, PTHrP, mainly expressed by the perichondrial cells and chondrocytes at the ends of the cartilage elements, is critical for maintaining the proliferation zone by activating a PP2A/HDAC4/MEF2 cascade through the PTHR1 receptor to prevent premature onset of hypertrophy[9–11]. The pace-keeping function of PTHrP is mediated through Ihh that is a necessary and direct inducer of PTHrP expression in the periarticular region[12–14]. As Ihh is only expressed in the prehypertrophic and early hypertrophic chondrocytes and thus suppressed by PTHrP signaling, it forms a negative feedback loop with PTHrP to ensure a proper rate of transition from proliferation to hypertrophy[15]. Besides the indirect regulation of hypertrophy, Ihh also directly promotes chondrocyte proliferation as well as the columnar organization of chondrocytes within the proliferating zone[14,16]. Thus, Ihh together with PTHrP represents a central mechanism for regulating cartilage development.

Bmp signaling is also known to modulate multiple aspects of skeletal development[17]. Studies have shown distinct expression patterns for different members of the Bmp family, including Bmp2, Bmp4 and Bmp5 in the perichondrium, Bmp2 and Bmp6 in the hypertrophic chondrocytes and Bmp7 in the proliferating chondrocytes[18]. Similarly, both type I and type II receptors for Bmp signaling are expressed with characteristic patterns within the growth plate. In particular, Bmpr1a is expressed highly in columnar, prehypertrophic and hypertrophic chondrocytes, coinciding with a high level of Smad1/5/8 phosphorylation, indicating active canonical Bmp signaling[18,19]. More importantly, deletion of Bmpr1a in chondrocytes impairs chondrocyte proliferation, column formation as well as hypertrophy, whereas additional removal of one Bmpr1b allele further exacerbates the defects[19]. Similarly, inducible deletion of Bmpr1a in the cartilage of newborn mice severely suppresses chondrocyte proliferation, proteoglycan production and hypertrophy in the growth plate[20]. Conversely, forced-expression of a constitutively active form of Bmpr1a (caBmpr1a) in chondrocytes appears to accelerate the transition from proliferation to hypertrophy[21]. However, the mechanisms through which Bmp signaling promotes chondrocyte proliferation and hypertrophy are not well understood.

Compared to the molecular regulators, much less is known about potential metabolic changes accompanying cartilage development. Although glucose is believed to be the main energy source as well as a major precursor for proteoglycan synthesis in chondrocytes, it is not known whether glucose uptake or metabolism is regulated across the different zones in the growth plate. An early study with a bioluminescence technique applied to sections of the chicken epiphyseal growth cartilage shows that glucose, glucose-6-phophate and lactate are present at the highest level in the hypertrophic chondrocytes, thus associating active glycolysis with hypertrophy[22]. However, a more recent study reports that inhibition of glycolysis in chondrocytic cell lines with NaF reduces intracellular ATP levels and causes hypertrophy-like changes[23]. Thus, the relationship between glycolysis and chondrocyte hypertrophy remains unclear. In addition, although multiple members of the Glut family have been implicated in glucose transport in chondrocytes, the physiological relevance of each transporter has not been demonstrated[24]. Finally, it is not clear whether and how growth factors modulate glucose metabolism in the developing cartilage. A recent study has indicated that Igf2 deletion leads to elevated and imbalanced glucose metabolism in the growth plate cartilage, but it remains unclear whether such dysregulation is directly due to the loss of Igf2 signaling or secondary to other changes[25]. Overall, further studies are necessary to understand the potential role and mechanism of metabolic regulation in skeletal development.

Here by genetic deletion we show that Glut1-mediated glucose metabolism is dispensable for the formation of the initial cartilage anlage, but is essential for the subsequent growth by supporting chondrocyte proliferation, cartilage matrix production as well as the transition to hypertrophy. Moreover, consistent with its known function in promoting chondrocyte proliferation and hypertrophy, Bmp signaling induces Glut1 transcription through mTORC1 and Hif1a in chondrocytes.

## Results

**Dynamic expression of Glut1 in developing cartilage.** As a first step to investigate the role of glucose metabolism in endochondral skeletal development, we examined the expression pattern of classic glucose transporters of the Glut family. RT-qPCR experiments showed that Glut1 mRNA level was two-three orders of magnitude higher than that of Glut2, Glut3, or Glut4 in primary chondrocytes. Previous work by others has documented that Glut1 mRNA was diffusely expressed throughout the limb mesenchyme prior to the onset of chondrogenesis in the mouse embryo, but was later upregulated in the cartilage primordia[26]. Therefore we focused on characterizing the expression of Glut1 protein by immunofluorescence staining. At E12.5 when skeletal elements began to form in the limbs of mouse embryos, Glut1 was notably enriched in the cartilage anlagen over the surrounding tissues (Fig. 1a, upper). At E14.5, Glut1 was elevated in the middle of the cartilage template, coinciding with the onset of chondrocyte hypertrophy (Fig. 1b, upper). By E16.5 and E18.5 when the growth plate cartilage was organized into discreet domains, Glut1 was most highly expressed in the prehypertrophic

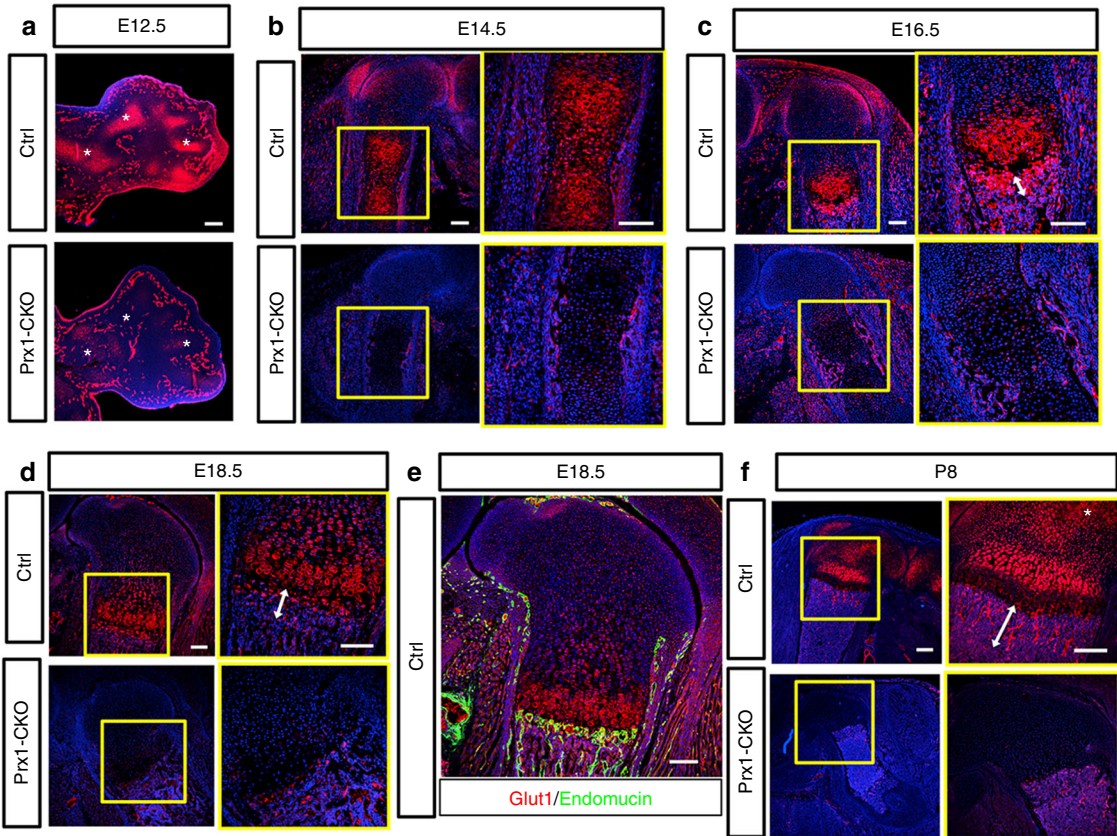

**Fig. 1** Genetic deletion with Prx1-Cre eliminates Glut1 in the long bones of the mouse. **a** Immunofluorescence staining for Glut1 on longitudinal sections of the forelimbs of E12.5 control (Ctrl) or mutant littermates (Prx1-CKO). Asterisk denotes cartilage elements. Note comparable expression in the vasculature between control and mutant embryos. **b**–**f** Glut1 immunostaining on longitudinal sections of the femur of littermate control or mutant embryos at E14.5 (**b**), E16.5 (**c**), E18.5 (**d**, **e**), and P8 (**f**). Boxed areas are shown at higher magnification to the right. Endomucin immunostaining in green marks endothelial cells (**e**). Double arrowheads denote trabecular bone region (**c**, **d**, **f**). Scale bar: 100 μm (**a**–**e**) or 200 μm (**f**). Each image is representative of three animals

and early hypertrophic regions, with apparent down-regulation in the late hypertrophic chondrocytes (Fig. 1c, d, upper). A strong Glut1 signal was also detected at the chondro-osseous junction (primary spongiosa) immediately below the hypertrophic zone (Fig. 1c, d, double-headed arrow). Co-staining with the endothelium marker endomucin showed that the blood vessels at the chondro-osseous junction expressed high levels of Glut1 (Fig. 1e). At postnatal day 8 (P8), strong Glut1 expression persisted in the early hypertrophic region and the primary spongiosa (Fig. 1f, upper). In addition, Glut1 was elevated in the presumptive secondary ossification center where chondrocytes were undergoing hypertrophy (Fig. 1f, asterisk). Throughout the stages examined, Glut1 expression in the perichondrium or bone collar was less prominent than that in the prehypertrophic and early hypertrophic cartilage. Overall, Glut1 levels greatly increase in chondrocytes during the early phases of hypertrophy.

**Dependence of skeletal growth on Glut1.** To explore the potential requirement of Glut1 in limb skeletal development, we used Prx1-Cre to delete *Glut1* in all of the limb mesenchyme including precursors of chondrocytes and osteoblasts. Besides the limbs, Prx1-Cre also targets the precursors for the calvarium or the sterna. Immunostaining confirmed that Glut1 protein was essentially eliminated from the limb cartilage (but not the limb vasculature) of the Prx1-Cre; *Glut1*<sup>f/f</sup> embryos (herein termed *Prx1-CKO*) by E12.5 (Fig. 1a, lower). Examination of the later stages (E14.5, E16.5, E18.5, P8) confirmed complete absence of Glut1 in the limb skeletal elements, whereas the expression in

endothelial cells (not targeted by Prx1-Cre) remained intact (Fig. 1b–d, f, lower). The mutant mice developed to term and survived postnatally but displayed obvious shortening of the limbs at birth. Whole-mount skeletal staining at E18.5 confirmed that all limb skeletal elements were notably shorter in the *Prx1-CKO* embryos than normal (Fig. 2a–c). The skull bones appeared to be normal but the sternum was under-mineralized in the mutant (Fig. 2d, e). The short limb phenotype was already obvious at E16.5 when the scapula and the humerus also exhibited hypomineralization, but both calvaria and sterna appeared normal at this stage (Fig. 2f–j). At E14.5, the limb skeletal elements appeared to have a normal length but were under-mineralized (Fig. 2k–m). Quantification confirmed that the femur and the humerus essentially stopped growing after E14.5 in the mutant (Fig. 2n, o). Remarkably, at E12.5 the cartilage anlagen in the mutant embryo were largely normal in shape and size despite efficient elimination of Glut1 protein by Prx1-Cre at this stage, as shown earlier (Fig. 2p-r). Thus, Glut1 in the limb mesenchyme appears to be dispensable for the initial chondrogenesis but is essential for the later skeletal growth in the mouse embryo.

**Role of Glut1 in chondrocyte proliferation and hypertrophy.** We next evaluated the cellular defects in the long bones of the *Glut1* mutant mice. Despite the relatively normal skeletal length at E14.5, H&E staining of the femur sections revealed a small but consistent reduction of the emerging hypertrophic zone (Supplementary Figure 1A). By E16.5, the mutant section lacked any recognizable hypertrophic chondrocytes or bone marrow, and the

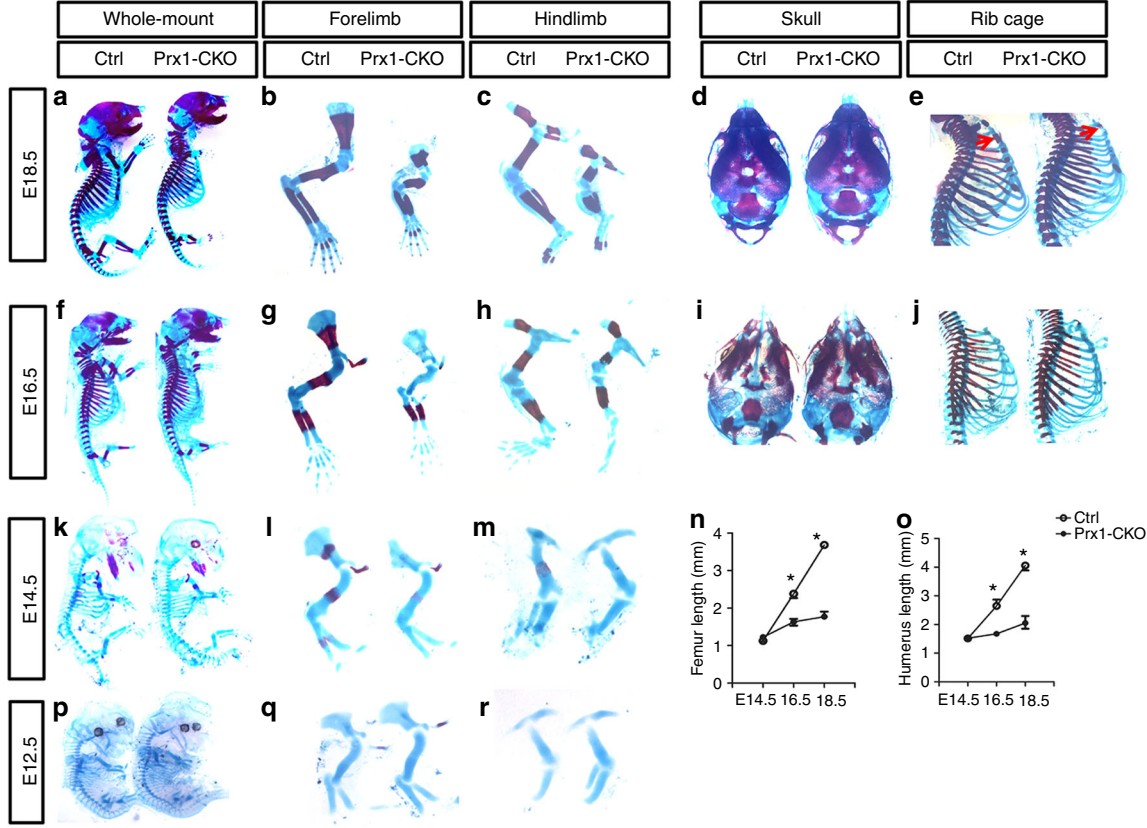

**Fig. 2** Deletion of Glut1 with Prx1-Cre leads to severe limb shortening. **a–m**, **p–r** Whole-mount skeletal staining of normal (Ctrl) or mutant (Prx1-CKO) littermate embryos at E18.5 (**a–e**), E16.5 (**f–j**), E14.5 (**k–m**), and E12.5 (**p–r**). Red arrows denote hypomineralization of the mutant sternum compared to control at E18.5. **n**, **o** Length measurements of femurs (**n**) and humeri (**o**) at indicated embryonic stages. *$p < 0.05$, $n = 3$, error bar indicates SD, two-way ANOVA and Bonferroni post-test. Each image is representative of three mice

presumptive marrow region was occupied by bony tissue that appeared to be ingrowth of the bone collar (Fig. 3a). In addition, the typical columnar organization of chondrocytes was entirely missing in the mutant cartilage. The same histological abnormalities were observed at E18.5 (Supplementary Figure 1B). Similarly, no columnar or hypertrophic chondrocytes of the normal morphology were observed in the mutant mouse at P8 (Fig. 3b, blue box). Furthermore, whereas normally at P8 a secondary hypertrophic zone formed within the epiphysis, prefiguring the secondary ossification center, no such hypertrophy occurred in the mutant cartilage (Fig. 3b, red box). As a result, no secondary ossification center developed in the mutant mouse at P16 (Fig. 3c). To confirm the apparent defect in chondrocyte hypertrophy, we performed immunofluorescence staining for collagen X (ColX), a marker for hypertrophic chondrocytes. The results showed the ColX-positive domain was much reduced at E14.5, and virtually absent at E16.6, E18.5, and P8 in the mutant (Fig. 3d). At P8, ColX was normally also evident in the secondary hypertrophy zone, but it was not detectable in the equivalent area in the mutant animal. MMP13, expressed by both terminal hypertrophic chondrocytes and osteoblasts, was absent in the cartilage of *Prx1-CKO* mice at either E16.5 or E18.5, even though it was detected in the osteoblasts at E18.5 (Fig. 3e). Therefore, both histology and molecular marker analyses demonstrate that Glut1 is essential for proper chondrocyte hypertrophy.

We then examined the potential effect of *Glut1* deletion on chondrocyte proliferation. To discern the nascent chondrocytes from the surrounding mesenchyme at E12.5, we performed immunostaining for Sox9, and confirmed that chondrogenesis was essentially normal at the molecular level even though Glut1

was virtually undetectable in the cartilage anlagen (Supplementary Figure 2A). Moreover, EdU labeling experiments indicated that *Glut1* deletion did not impair the proliferation rate of Sox9+ cells at this stage (Supplementary Figure 2B, C). However, *Glut1* deletion caused a marked reduction in EdU labeling throughout the proliferative zone of the femur at E14.5, E16.5, and E18.5 (Fig. 4a, Supplementary Figure 2D, E). Quantification within the columnar region confirmed that the percentage of EdU+ chondrocytes was reduced by ~50% at E14.5, and ~80% at E16.5 or E18.5 in the absence of Glut1 (Fig. 4b). On the other hand, TUNEL assays detected no obvious apoptosis in the cartilage upon *Glut1* deletion at any of the stages (Fig. 4c, Supplementary Figure 2D, E). In fact, for reasons unknown at-present, apoptosis normally detected at the interface of the cartilage and the diaphyseal perichondrium appeared to be suppressed in the mutant embryo (Fig. 4c, Supplementary Figure 2D, E). Thus, Glut1 is critical for the proper proliferation of growth plate chondrocytes but dispensable for their survival.

As a major function of chondrocytes is to produce extracellular matrix proteins, we next asked whether Glut1 is required for this process. Safranin O staining consistently detected a lower intensity in the Glut1-deficient cartilage at all stages examined, suggesting a general decrease in proteoglycan levels (Fig. 4d, Supplementary Figure 3). Quantification of mRNA in the growth plate cartilage with RT-PCR revealed that *Glut1* deletion markedly reduced the expression of *Col2a1* and the proteoglycan genes *aggrecan* (*Acan*), *lumican* (*Lum*), and *epiphycan* (*Epyc*) (Fig. 4e). Therefore, Glut1 is critical for supporting the expression of cartilage matrix genes by chondrocytes.

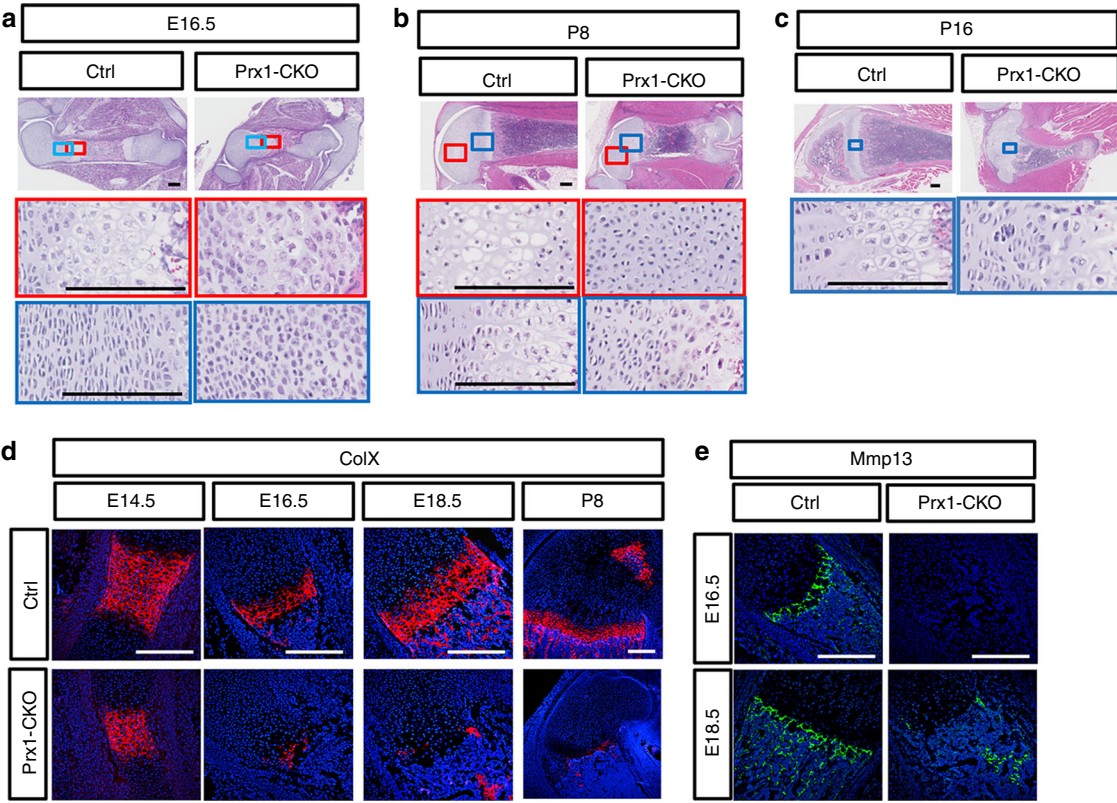

**Fig. 3** Glut1 is indispensable for proper maturation of chondrocytes. **a–e** Histology of the femur at E16.5 (**a**), p8 (**b**), and p16 (**c**). Boxed areas shown at higher magnification below. **d**, **e** Immunofluorescence staining of ColX (**d**) or Mmp13 (**e**). Scale bar: 200 μm. Each image is representative of three mice

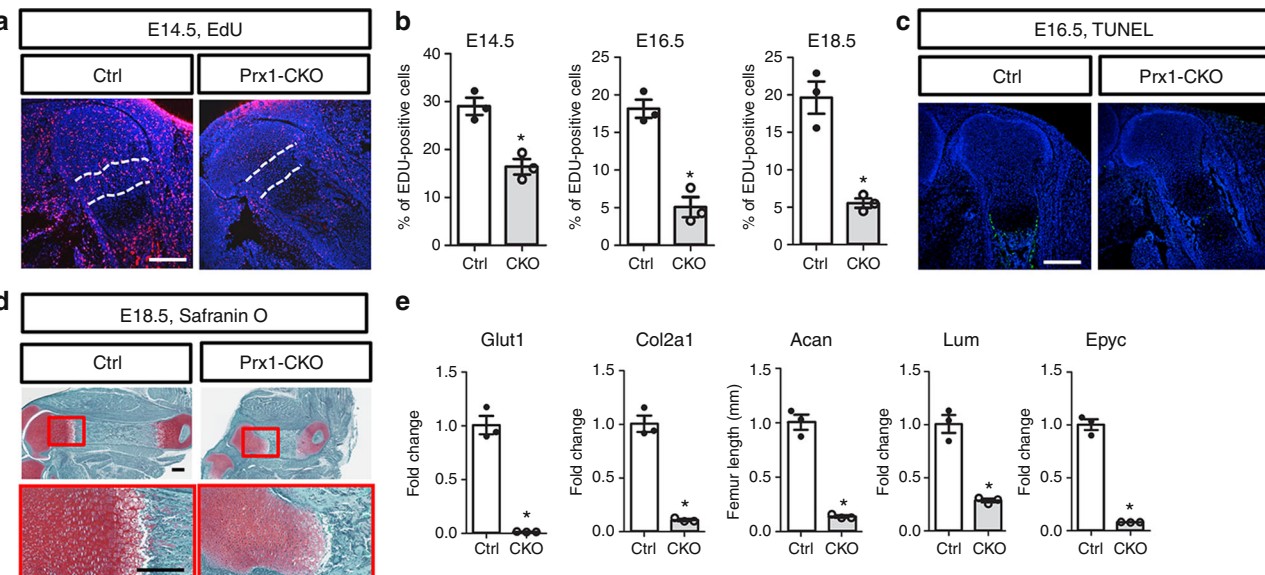

**Fig. 4** Glut1 is necessary for chondrocyte proliferation and matrix production. **a** Representative images of EdU labeling (red) in the femur at E16.5. Region between dashed lines is used for quantification of labeling index. **b** EdU quantification. *$p < 0.05$, $n = 3$ mice, error bar indicates SD, two-tailed Student's t-test. Scale bar: 250 μm. **c**, **d** Representative images of TUNEL (**c**) or Safranin O staining (**d**). **e** Gene expression assayed by RT-qPCR in cartilage. *$p < 0.05$, $n = 3$ mice, error bar indicates SD, two-tailed Student's t-test. Scale bar: 250 μm. Images are representative of three mice

Since previous studies have implicated Glut1 in osteoblast differentiation, we examined bone formation in the *Prx1-CKO* embryos[26]. Immunofluorescence staining of Sp7 detected pre-osteoblasts and osteoblasts in both the perichondrium/periosteum and the primary ossification center of wild type embryos at E16.5 (Fig. 5a, left). In the *Prx1-CKO* littermate, although a bone marrow cavity was absent, preosteoblasts and osteoblasts were abundant at the perichondrium/periosteum and have begun to invade the center of the cartilage template (Fig. 5a, right). Immunostaining of endomucin indicated that blood vessels were abundant in the thickened bone collar but were just beginning to invade the cartilage proper (Fig. 5b). Staining of type I collagen

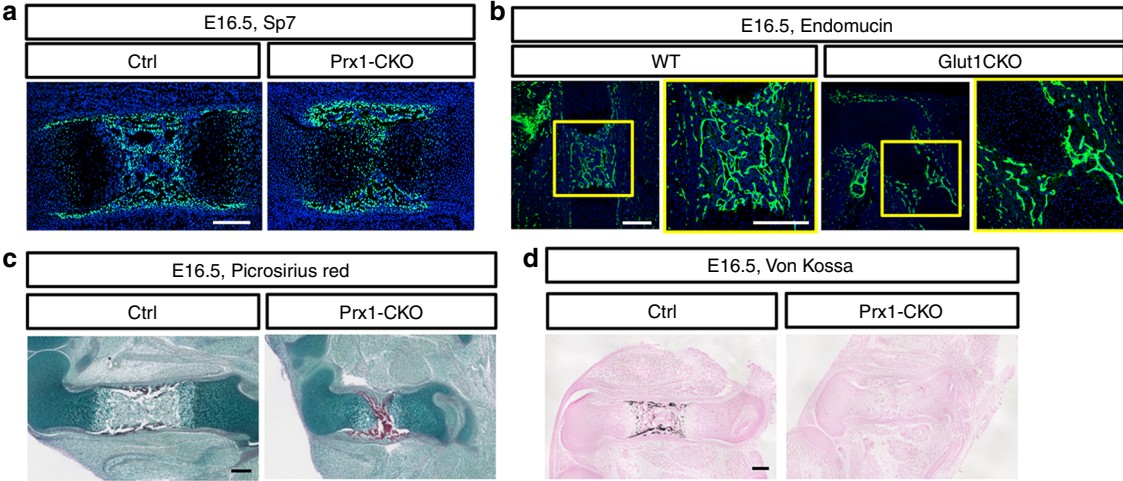

**Fig. 5** Loss of Glut1 impedes mineralization of bone matrix at E16.5. **a**, **b** Immunofluorescence staining of Sp7 (**a**) or Endomucin (**b**) on longitudinal sections of the femur. **c** Picrosirius Red staining for collagen I. **d** von Kossa staining for minerals. Scale bar: 200 μm. Each image is representative of three mice

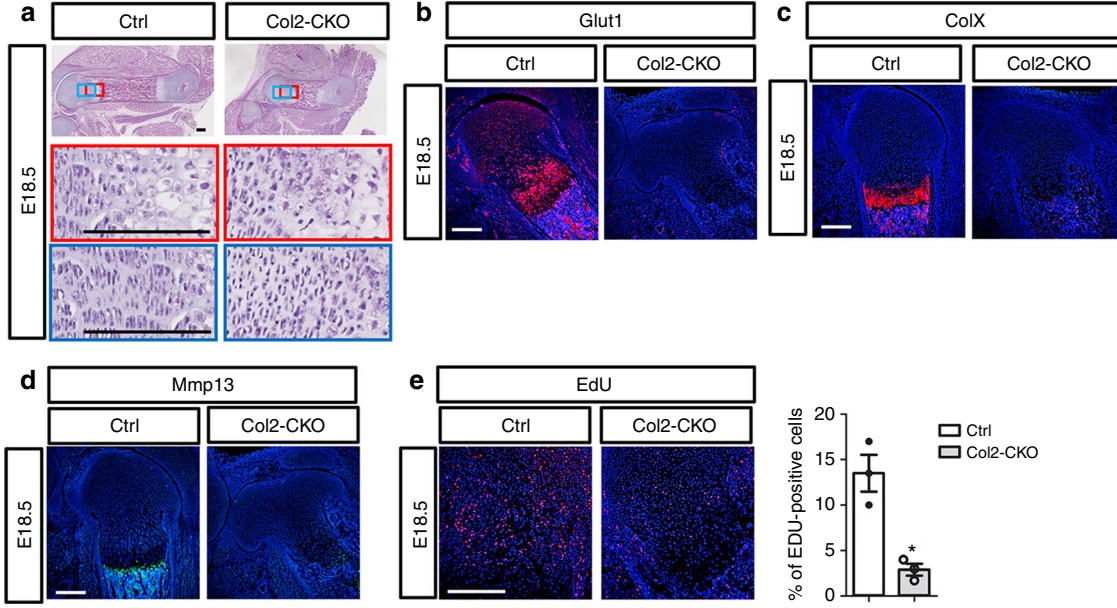

**Fig. 6** Deletion of Glut1 with Col2-Cre impairs chondrocyte proliferation and maturation. **a** H&E staining of longitudinal femoral sections from normal (Ctrl) versus Col2-Cre; Glut1^f/f (Col2-CKO) littermate embryos at E18.5. **b–d** Immunofluorescence staining of Glut1 (**b**), ColX (**c**) or Mmp13 (**d**) on E18.5 femoral sections. **e** EdU labeling of chondrocytes in the femur at E18.5. Signal is in red. *$p < 0.05$, $n = 3$ mice, error bar indicates SD, two-tailed Student's $t$-test. Scale bar: 200 μm

with Picrosirius Red confirmed that the osteoblasts were actively producing bone matrix in the mutant mice (Fig. 5c). However, von Kossa staining revealed that mineralization of the bone matrix was undetectable in the femur of two out of three *Prx1-CKO* embryos analyzed at E16.5 (Fig. 5d). On the other hand, bone mineralization was readily detectable by E18.5 in the mutant, indicating only a modest delay in bone mineralization caused by *Glut1* deletion. Therefore, Glut1 is dispensable for osteoblast differentiation but appears to play a role in supporting the normal mineralizing activity of osteoblasts.

The use of *Prx1-Cre* does not distinguish the direct requirement for Glut1 in chondrocytes from potential indirect effects from the other mesenchymal cell types in the limb. To overcome this limitation, we deleted *Glut1* with *Col2-Cre* that targets both chondrocytes and osteoblasts but not the other limb mesenchymal cells. Similar to the *Prx1-CKO* mutants, the *Col2-Cre; Glut1^f/f* mice (*Col2-CKO*) exhibited severe shortening of the limb

skeletal elements coupled with no obvious columnar or hypertrophic chondrocytes at E18.5 (Fig. 6a). Immunostaining confirmed that Glut1 was undetectable in the cartilage of *Col2-CKO* embryos (Fig. 6b). Both the pan-hypertrophy marker ColX and the terminal hypertrophy marker Mmp13 were absent in the mutant cartilage (Fig. 6c, d). Moreover, chondrocyte proliferation, as assayed by EdU labeling, was reduced by ~80% in the *Col2-CKO* embryo (Fig. 6e). These results therefore support the notion that Glut1 is directly required in chondrocytes for their proper proliferation and hypertrophy.

**Induction of Glut1 by Bmp2-mTORC1-Hif1a signaling.** As Bmp signaling is known to promote chondrocyte proliferation and hypertrophy, we next tested the hypothesis that Bmp signaling and Glut1 may be linked in the regulation. In keeping with previous findings, we found that deletion of *Bmpr1a* with *Col2-*

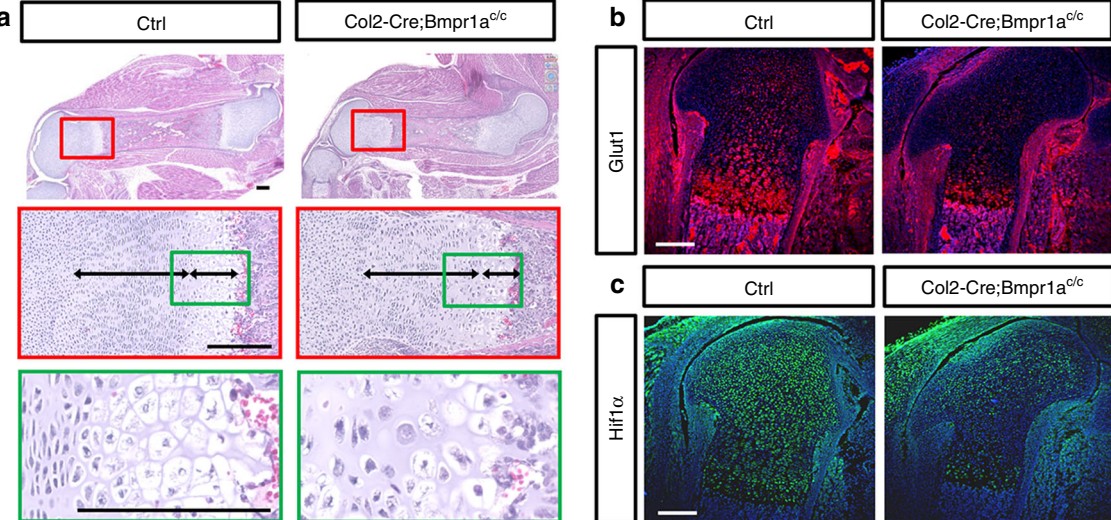

**Fig. 7** Deletion of Bmpr1a with Col2-Cre disrupts chondrocyte maturation and reduces Glut1 expression. **a** Representative H&E staining of the femur from normal (Ctrl) versus Col2-Cre; Bmpr1a$^{f/f}$ littermate embryos at E18.5. Boxed areas shown at high magnification below. **b**, **c** Immunofluorescence staining of Glut1 (**b**) or Hif1a (**c**) in the femur at E18.5. Three littermate pairs were analyzed with similar results. Scale bar: 200 μm. Each image is representative of three mice

*Cre* resulted in shorter skeletal elements associated with disorganization of the columnar region and a diminished hypertrophic zone (Fig. 7a)[19]. Although the overall limb shortening phenotype here was less severe, the defect in chondrocyte maturation was qualitatively similar to that caused by *Glut1* deletion. Importantly, immunofluorescence staining showed that Glut1 levels were markedly reduced in the columnar and hypertrophic chondrocytes of the *Col2-Cre; Bmpr1a*$^{f/f}$ embryos (Fig. 7b). In addition, Hif1a, a known transcriptional activator for *Glut1*, was also notably diminished in the columnar and hypertrophic cartilage in the absence of Bmpr1a (Fig. 7c). Thus, Bmp signaling appears to function upstream of Glut1 in the developing growth plate cartilage.

We then investigated the possibility that Bmp signaling might directly modulate Glut1 expression. Treatment of primary chondrocytes with recombinant BMP2 increased the protein level of Glut1 by approximately twofold after 24 h, whereas recombinant Igf1 or the Hh agonist purmorphamine, both also known to regulate chondrocyte development, did not affect Glut1 levels (Fig. 8a). The BMP2 effect was time-dependent, as 6 h of treatment did not increase Glut1 protein levels (Supplementary Figure 4). Analyses of mRNA with RT-qPCR confirmed that *Glut1* was the predominant isoform over *Glut2, 3,* and *4* in chondrocytes, and that BMP2 specifically induced *Glut1* mRNA by over twofold after 24 h (Fig. 8b). Consistent with the induction of Glut1, BMP2 stimulated glucose consumption by the chondrocytes after 48 h (Fig. 8c). Thus, Bmp signaling induces Glut1 expression and promotes glucose metabolism in primary chondrocytes.

To elucidate the molecular mechanism responsible for Glut1 induction, we examined the role of Smad signaling downstream of Bmp. Knockdown of Smad4 with two different shRNA notably reduced the induction of Glut1 protein and mRNA by BMP2 (Fig. 8d, e). Although the result in itself suggests that the Smad transcription factors might directly activate *Glut1* gene transcription in response to Bmp, a previous ChIP-seq study failed to detect Smad4 binding in the promoter region (−5kb to +5b around transcription start site) of the *Glut1* gene in the embryonic cartilage[27]. We therefore investigated the potential role of mTORC1 signaling due to its known relevance in cellular metabolism and its implication in Bmp signaling[28]. After 24 h of treatment, BMP2 induced the phosphorylated fraction of S6 (P-S6, a readout for mTORC1 activity) over the total S6 levels, but the induction was diminished upon Smad4 knockdown (Fig. 8f). As previous studies have shown that Smad4 reduces *Pten* mRNA which encodes a negative regulator of mTORC1 activity, we investigated Pten suppression as a potential mechanism for Bmp to activate mTORC1[29,30]. Indeed, BMP2 reduced *Pten* mRNA in chondrocytes and the suppression was abolished upon Smad4 knockdown (Fig. 8g). In support of the functional importance of mTORC1, inhibition of mTORC1 signaling with either rapamycin or Torin1 abolished the induction of *Glut1* by BMP2 while also reducing the basal level of Glut1 (Fig. 9a). Similarly, the induction of Glut1 mRNA was either completely abolished or markedly suppressed by the mTOR inhibitors (Fig. 9b). Therefore, Bmp signaling stimulates Glut1 expression likely downstream of Smad4 and mTORC1 activation.

We next explored the mechanism downstream of mTORC1 leading to Glut1 induction. As shown earlier, Hif1a was diminished concurrently with Glut1 in the cartilage of *Col2-Cre; Bmpr1a*$^{c/c}$ embryos (Fig. 7b, c). As a previous study has reported that mTORC1 enhances the translation of Hif1a, we tested Hif1a as a potential mediator of Bmp signaling in activating Glut1 expression[31]. BMP2 consistently induced Hif1a protein in primary chondrocytes after 24 h, but the induction was abolished by either rapamycin or Torin1 (Fig. 9c). Moreover, chemical inhibition of Hif1a with either PX-478 or acriflavine dose-dependently eliminated the induction of Glut1 by BMP2 (Fig. 9d). Collectively, the results demonstrate that Bmp signaling upregulates Glut1 through activation of mTORC1 and Hif1a in chondrocytes.

## Discussion

The present study provides genetic evidence, that the Glut1 glucose transporter is required for specific aspects of endochondral skeletal development. Deletion of *Glut1* in the pre-chondrogenic limb mesenchyme with *Prx1-Cre* does not interfere with skeletal

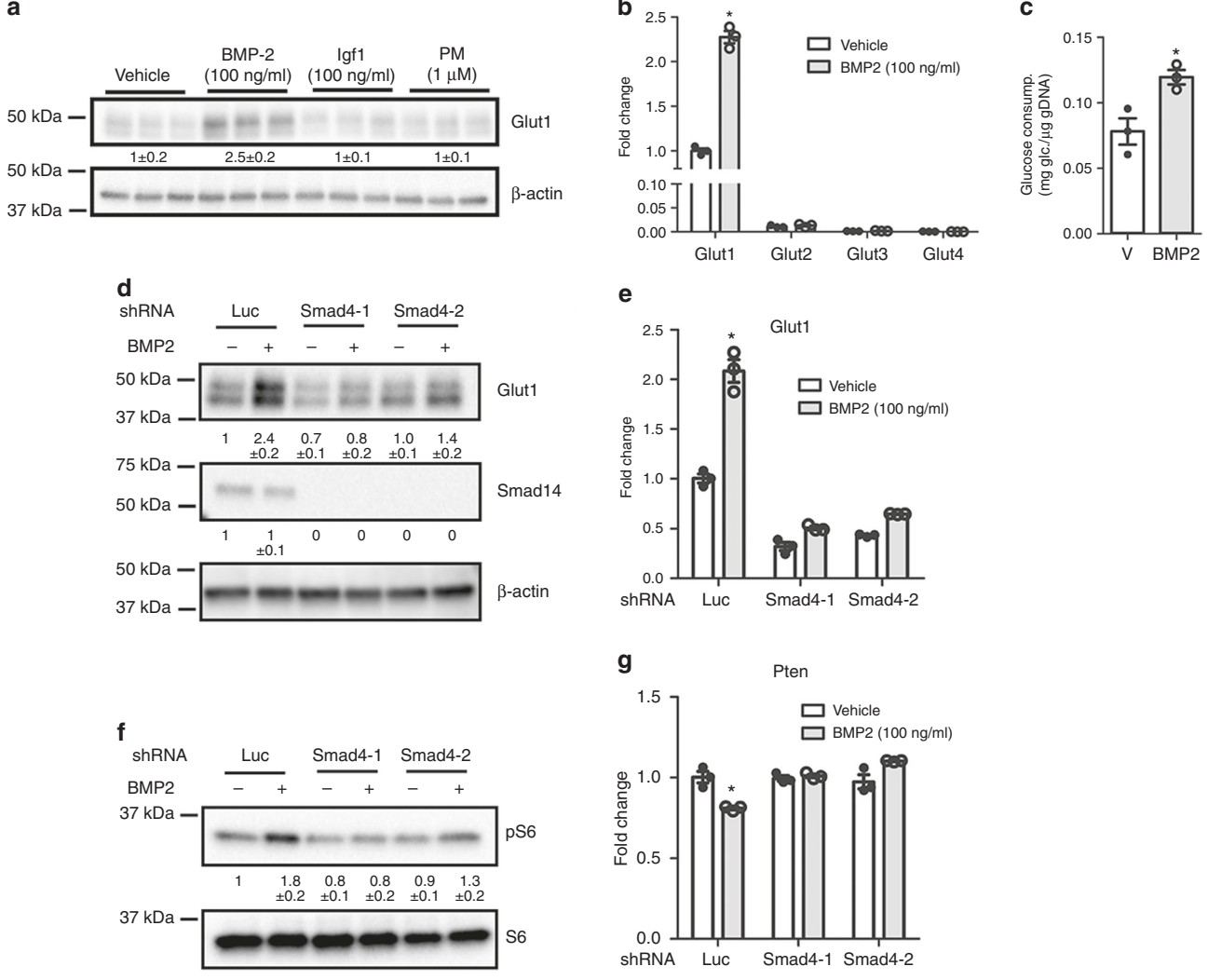

**Fig. 8** BMP2 induces Glut1 expression via Smad4 and mTORC1. **a** Western blot analyses of primary chondrocytes after 24 h of treatment as indicated. **b** RT-qPCR analyses in primary chondrocytes with or without BMP2 treatment for 24 h. *$p < 0.05$, $n = 3$, error bar indicates SD, two-tailed Student's $t$-test. **c** Glucose consumption by primary chondrocytes with or without BMP2 treatment for 48 h. Data normalized to genomic DNA. *$p < 0.05$, $n = 3$, error bar indicates SD, two-tailed Student's $t$-test. **d, e** Effects of Smad4 knockdown on Glut1 induction by BMP2 assayed by western blot (**d**) or RT-qPCR (**e**). shLuc used as negative control. Two different Smad4 shRNA used. *$p < 0.05$, $n = 3$, error bar indicates SD, two-tailed Student's $t$-test. **f** Western blot analyses of Smad4 knockdown on BMP2-induced mTORC1 activation. **g** RT-qPCR analyses of Pten mRNA. *$p < 0.05$, $n = 3$, error bar indicates SD, two-tailed Student's $t$-test. Quantification of all western blots denotes average fold change over vehicle control after normalization to β-actin (**a, d**) or total S6 (**f**) (±SD, $n = 3$)

patterning or the initial expansion of the cartilage anlagen, but essentially halts the subsequent longitudinal growth by impeding chondrocyte proliferation and hypertrophy. *Glut1* removal in the cartilage anlage with *Col2-Cre* causes the same proliferation and hypertrophy defects, thus arguing for a direct requirement for Glut1 in chondrocytes. The study further identifies Glut1 as a downstream effector of Bmp signaling in cartilage development, as genetic deletion of *Bmpr1a* diminishes Glut1 expression in the growth plate whereas BMP2 induces *Glut1* transcription via mTORC1-Hif1a signaling in chondrocytes. The work therefore highlights the coordination between growth factor signaling and glucose metabolism during cartilage development.

Future studies are necessary to understand the biochemical basis for glucose metabolism to control chondrocyte proliferation and hypertrophy. The genetic roles uncovered here are consistent with the modest yet widespread expression of Glut1 throughout the growth plate, and a marked upregulation in the columnar and early hypertrophic zones. Besides providing energy, glucose supplies the carbon skeleton for biomass syntheses in the cell[32].

In the proliferating chondrocytes, glucose metabolism likely generates the intermediate metabolites necessary for de novo synthesis of nucleotides and lipids in support of cell proliferation[33]. The upregulation of Glut1 in the columnar chondrocytes likely reflects an increase in biosynthesis as those cells are not only highly proliferative but also produce large quantities of extracellular matrix proteins. Similarly, increased glucose metabolism in the early hypertrophic chondrocytes corresponds to the enhanced production of cellular dry mass[3]. The intensified biosynthesis in both columnar and early hypertrophic chondrocytes is consistent with a strong mTORC1 activity in those cells[34]. Thus, glucose metabolism as reflected by Glut1 expression closely tracks with the biosynthetic activity of chondrocytes in the developing growth plate.

It is somewhat unexpected that osteoblast differentiation appears to be largely normal upon the deletion of *Glut1*. Previous work has shown that glucose is a major nutrient for osteoblasts and that both Wnt and PTH induce osteoblast differentiation and activity partly through stimulating glycolysis[35–39]. Moreover,

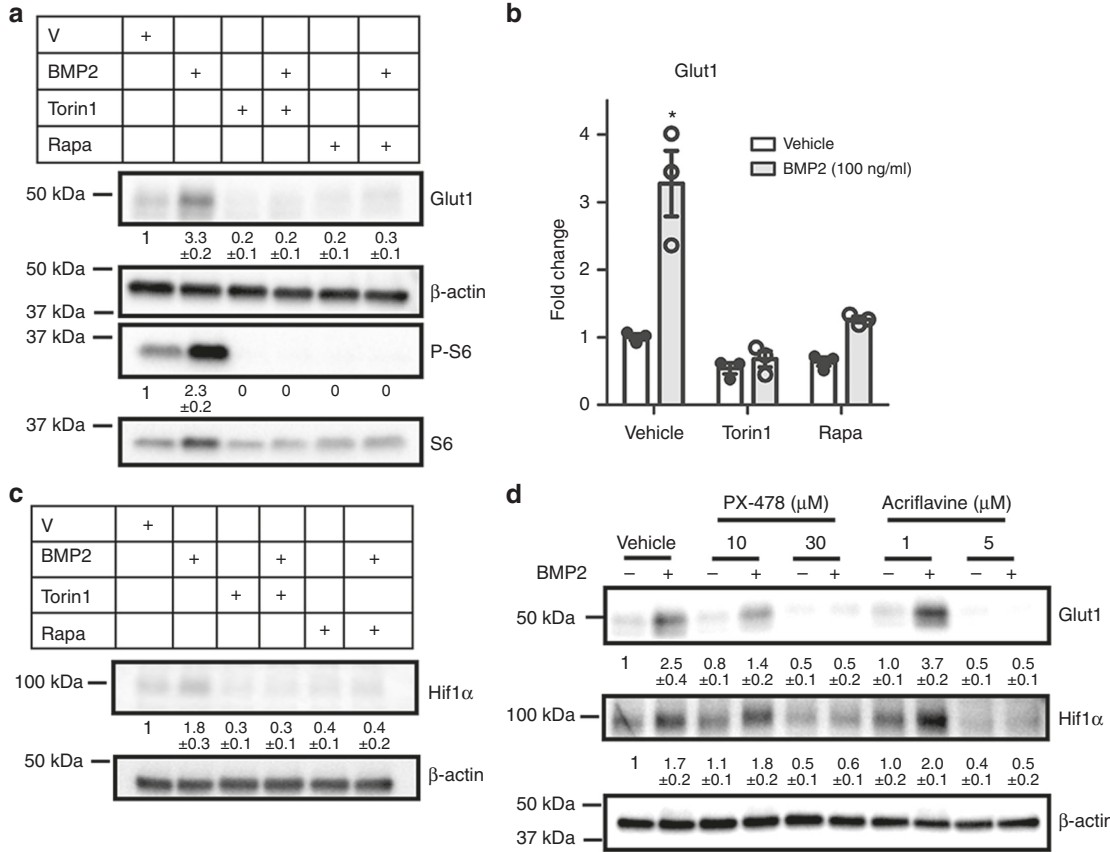

**Fig. 9** Hif1α mediates Glut1 induction downstream of Bmp-mTORC1 signaling. **a–c** Western blot (**a**, **c**) or RT-qPCR (**b**) analyses in primary chondrocytes after 24 h of BMP2 treatment with or without mTOR inhibitors. Rapa: rapamycin. **d** Western blot analyses in primary chondrocytes in response to BMP2 with or without the Hif1α inhibitors for 24 h. Quantification of all western blots denotes average fold change over vehicle control after normalization to β-actin (±SD, $n = 3$). *$p < 0.05$, $n = 3$, error bar indicates SD, two-tailed Student's $t$-test

Glut1 is the main glucose transporter in osteoblast lineage cells and has been shown to engage in a feedforward mechanism with Runx2 to control the onset of osteoblast differentiation from the progenitors[26]. Although we noted a lack of bone collar staining by Alizarin Red in the limbs of Prx1-CKO embryos at E14.5, we could not discern potential direct versus indirect effects on the osteoblast lineage due to the concurrent defect in chondrocyte hypertrophy. In any case, by E16.5 functional osteoblasts were abundant in the bone collar, as indicated by Osx expression and collagen I deposition. Thus, any requirement for Glut1 in osteoblast differentiation appears to be transient. However, as von Kossa staining detected a delay in bone matrix mineralization in the E16.5 *Prx1-CKO* embryos, Glut1 may be important for supporting the full range of osteoblast activity. The relative normalcy of osteoblast differentiation in the absence of Glut1 may indicate functional redundancy with the other glucose transporters. However, we cannot rule out metabolic plasticity that may allow the cells to switch to other nutrient substrates to support differentiation when glucose metabolism is compromised.

Bmp signaling likely regulates cartilage development through multiple mechanisms. Deletion of both *Smad1* and *Smad5* in chondrocytes, similar to the double deletion of *Bmpr1a* and *Bmpr1b*, results in the absence of most endochondral skeletal elements, thus indicating that Smad1/5 are likely critical mediators of Bmp signaling in cartilage development[40,41]. Studies of the cartilage remnants in the *Smad1/5* double mutants have uncovered profound defects in chondrocyte proliferation and hypertrophy[41]. Although Smad1/5 have been shown to activate the transcription of *Col10a1*, a marker gene for chondrocyte

hypertrophy, the Smad target genes responsible for the hypertrophic program remain unknown[42]. Here we show that Smad4 at least partially mediates Bmp function in activating mTORC1 and Hif1a to enhance *Glut1* transcription and glucose metabolism in support of hypertrophy. Suppression of *Pten* appears to link Bmp-Smad4 signaling to the activation of mTORC1, but how Smad4 reduces *Pten* mRNA is unknown. Furthermore, additional further studies are warranted to determine whether loss of Smad1/5 increases *Pten* and activates mTORC1 and glycolysis in the cartilage in vivo. Finally, additional mechanisms beyond Smad signaling may also contribute to mTORC1 activation in response to Bmp.

The finding about the critical dependence of chondrocyte hypertrophy on Glut1 may have practical implications. Although articular cartilage normally maintains a stable state without progressing through the maturation process as seen in the growth plate, hypertrophy of articular chondrocytes commonly occurs in osteoarthritis (OA) and is believed to contribute to the pathogenesis[43]. Thus, suppressing Glut1 or other components of the glycolytic pathway to prevent the abnormal hypertrophy may help slowing down the progression of OA. In addition, current efforts with articular cartilage replacement therapy are hampered by unwanted progression of tissue-engineered cartilage towards hypertrophy[44]. Therefore, inhibiting Glut1 activity might help creating permanent hyaline cartilage in vitro for tissue therapy.

## Methods
**Mouse strains**. Prx1-Cre[45], Col2-Cre[46], Glut1[f/f 47], and Bmpr1a[f/f 48] mouse lines have been previously reported. The Animal Studies Committee at Washington University approved all mouse experiments.

**Primary chondrocyte cultures**. Primary chondrocytes were isolated from the rib cages of 3-day-old C56BL6/J mice, and cultured in DMEM supplemented with 10% FBS according to a published method[49]. Briefly, the rib cages were dissected free of soft tissues in cold PBS before being further cleaned up with 2 mg/ml of protease (Sigma Aldrich, P6911I) in PBS and then 3 mg/ml of collagenase (Sigma Aldrich, P6885) in PBS, each for 15 min at 37 °C with agitation. After three rinses with PBS, the rib cages were digested with 0.3 mg/ml of collagenase in DMEM (Gibco,11965) for overnight at 37 °C with 5% $CO_2$. Primary chondrocytes were dissociated by pipetting and passing through a 70 µm cell strainer (BD Falcon, 08-771-2), and finally cultured in DMEM with 10% FBS. Subsequently, $2.5 \times 10^5$ cells were seeded into each well of 24-well plates and grown to 80-90% confluence before being treated with 100 ng/ml BMP2 (R&D Systems, 355-BM-010), 100 ng/ml IGF1 (R&D Systems, 791-MG-050) or 1 µM Purmorphamine (Millipore, 540223) for indicated time. In certain experiments, BMP2 was used together with 100 nM Torin1 (Selleckchem, S2827) or 20 nM Rapamycin (Selleckchem, S1039) for 24 h. Glucose consumption was measured with Glucose (HK) Assay Kit (Sigma Aldrich, GAHK20). For shRNA knockdown experiments, Lentiviruses expressing shRNA were produced by transfecting HEK293T cells with MISSION shRNA plasmids targeting Smad4 (Sigma-Aldrich, TRCN0000025885, TRCN0000345739) or luciferase as a negative control, together with the packaging plasmids pMD2.G and psPAX2 by using the Fugen6 transfection reagent (Promega). The viral media was added to the cells for 8 h before being replaced with fresh growth media for additional 48 h. When BMP2 was used, the cells were then treated with BMP2 for 24 h before being harvested.

**Analyses of mouse embryos**. Both male and female embryos were used without distinction in all analyses. Whole-mount embryonic skeleton was stained with Alizarin Red/Alcian Blue according to a published method[50]. Briefly, the embryos were skinned, eviscerated, and fixed in 95% ethanol for overnight and then changed to acetone for 24 h. The embryos were then stained with Alcian Blue–Alizarin Red staining solution containing 0.03% Alcian blue (Sigma Aldrich, A5268), 0.005% Alizarin Red (Sigma Aldrich, A5533), 10% Glacial acetic acid and 80 % ethanol for 3 days. Subsequently embryos were transferred to 95% Ethanol for 4 h, and cleaned up with 1% KOH (E16.5 and E18.5 embryos), or 0.5% KOH (E12.5 and E14.5), for 3 to 5 days. The embryos were cleared in 1 or 0.5% KOH plus 20% glycerol for 24 h, and subsequently in 1 or 0.5% KOH plus 50% glycerol for 24 h. The cleared skeletons were stored in 1 or 0.5% KOH with 80% glycerol.

For analyses on sections, embryonic limbs were dissected out in PBS, fixed in 10% neutral buffed formalin overnight at room temperature. For embryos older than E16.5, the limbs were either decalcified or non-decalcified (see below) before being processed for paraffin sectioning at 6 µm thickness. For von Kossa staining, the limbs were non-decalcified and the sections were stained with 2% silver nitrate and counterstained with Nuclear Fast red. For H&E, Alcian blue/Picrosirius Red, or Safarin O staining, the limbs were decalcified for 3 days with 14% EDTA (pH 7.2) prior to processing for paraffin sectioning.

**Fluorescence staining**. Pregnant female mice were injected intraperitoneally with EdU (Thermo Fisher Scientific, C10339) at 10 µg/g body weight 2 h before harvest. Embryonic limbs were dissected out, fixed with 4% paraformaldehyde (EMS, 15710) overnight, decalcified for 3 days with daily changes of 14% EDTA (pH 7.2), and incubated in 30% sucrose overnight for cryoprotection prior to embedding in optimal cutting temperature (OCT) (Tissue-Tek, 4583). Sections of 10 µm in thickness were obtained with a Leica cryostat equipped with Cryojane (Leica). EdU was detected by a Click reaction performed according to the manufacturer's instructions. TUNEL assay was performed with the In-Situ Cell Death Detection Kit Fluorescein (Roche, 11684795910). For immunostaining, the frozen sections were stained with specific antibodies against Glut1 (Santa Cruz, SC-7903), Endomucin (Santa Cruz, SC-65495), Osterix (Abcam, ab22552), Mmp13 (Abcam, ab39012), or Collagen type X (DSHB Hybridoma Product, X-AC9). The secondary antibodies were as follows: Alexa Fluor Alexa Fluor 488 goat anti-rabbit IgG (Life Science, A-11034), Alexa Fluor 594 goat anti-rabbit IgG (Life Science, R37117). Stained section were mounted with VECTASHIELD mounting medium containing DAPI (Vector Laboratories, H-1200).

**Western blots**. Total protein extracts were prepared from primary chondrocytes using RIPA buffer (Cell signaling, 9806) containing proteinase inhibitors and phosphatase inhibitors (Thermo Fisher Scientific, 78442). 10 µg of total protein was resolved by electrophoresis on Mini-PROTEAN® TGX™ Precast Gel (Bio-Rad), transferred onto a 0.45-µm pore PVDF Immobilon-P membrane (Millipore), and detected with a specific antibody against S6 (Cell Signaling, 2215), P-S6(S240/244) (Cell Signaling, 9452), β-Actin (Cell Signaling, 4970), Glut1 (Santa Cruz, SC-7903), or Hif1α (Novus, NB100-479). Membranes were blocked for 1 h at room temperature in 5% BSA (Roche) in TBS containing 0.1% Tween-20 (TBS-T) before overnight incubation with the primary antibody at 4 °C. The membranes were then washed with TBS-T, and further incubated at room temperature for 1.5 h with a HRP-conjugated anti-rabbit (GE Healthcare, NA934V) or anti-mouse (Cell Signaling, 7076S) secondary antibody. All blots were developed by using the Clarity Substrate Kit (Bio-Rad). Gel images were captured with Chemidoc (Bio-Rad). Each

experiment was repeated with a minimum of three independently prepared protein samples. The uncropped blot images are provided in Supplementary Figure 5.

**RT-qPCR**. Total RNA was isolated from primary chondrocyte cultures or embryonic epiphyseal cartilage with RNeasy mini kit (Qiagen, 74104). The epiphyseal cartilage was dissected from both forelimbs and the hindlimbs, treated with 0.25% trypsin EDTA for 15 min at 37 °C to remove the surrounding connective tissues, and then rinsed with Hank's Balanced Salt Solution (HBSS). Reverse transcription was performed using 500 ng of total RNA with iScript™ reverse transcription super mix (Bio-Rad, 1708891). qPCR reactions were set up in a 96-well format on an ABI StepOne Plus using SsoAdvanced™ universal CYBR Green Supermix (Bio-Rad, 1725274). β-actin was used for normalization in most cases unless otherwise indicated, and relative expression was calculated by the $2^{-(\Delta\Delta CT)}$ method.

## Data availability
The authors declare that all data supporting the findings of this study are available within the Article and its Supplementary Information files or from the corresponding author upon reasonable request.

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

## Acknowledgements

The work is supported by NIH grants AR055923, AR060456, and DK111212 (F.L.).

## Author contributions

S.-Y.L. conducted all experiments, prepared the figures, and helped with writing. E.D.A. provided the Slc2a1 floxed mouse and reviewed the manuscript. F.L. directed the study and wrote the paper.

## Additional information

**Competing interests:** The authors declare no competing interests.

