## [Peer Review File · Nature Communications]

Reviewers' Comments:

Reviewer #1:

Remarks to the Author:

In this study, Lee and colleagues report the novel finding that the glucose transported Glut1 is essential for growth plate development and chondrocyte biology downstream of BMP2/mTOR/HIF-1alpha. The authors use an elegant combination of in vivo and in vitro approaches for the testing of their hypothesis.

The paper is novel and important as it uncovers for the first time an unknown connection between BMP signaling and glucose metabolism in endochondral bone development. The data are solid and convincing. The quality of the figures is outstanding. The authors' conclusions are fully supported by the data as shown.

Minor point

Figure 7: Quantification of Western blot data should be provided.

Reviewer #2:

Remarks to the Author:

A rapid survey of the literature indicates that the field of the metabolic control of cell differentiation during skeletogenesis was open up recently by Wei et al. who looked at the role of glucose uptake in chondrocyte and osteoblast progenitor cells from E 10.5 onwards.

Following the path of this initial study Lee et al., focus this study on chondrocytes. As expected from this group the image are dazzling, the text is also well written. There are however several concerns with this paper that could be somewhat shorter in view of what has been published already. Here are some important points that need to be addressed.

- The expression of Glut1 in chondrocytes starting at E10.5, i.e., much earlier than in this paper, and with many more molecular markers has already been published (Wei et al., Figure 1) it is disconcerting that this is not acknowledged by the authors when they describe their own expression study/ Is there a reason for this omission?
- If this study has to be repeated it should be repeated at earlier stage of cell differentiation when Prx-Cre will be active, i.e., before E13.5. It should include molecular makers of chondrocyte differentiation not only structural genes, e.g., Sox9, Sox5, Sox6.
- A conceptual flaw of this study is that the analysis performed by the authors starts at E14.5 in Figure 2 and contains only one panel at E12.5 in Figure 1. This is a problem for two reasons, the first one is that that stage and in agreement with the pattern of expression of Prx itself there is already phenotypic abnormalities and so the reader is left with an unanswered question: Until when development happens normally? The second reason is that Prx-Cre is not really active beyond E14.5 and all we looking at are consequences of events occurring earlier.
- The authors state strongly that osteoblast differentiation is not affected in their model and yet a few lines later they acknowledge, rightly so, that Prx-cre may act indirectly and that there are limitations wit this mouse. So much so that they are forced to use other Cre drivers. In fact osteoblast differentiation when it is studied, is affected in this mouse model since Type I collagen expression is abolished but this point is ignored for reasons that are unclear.
- In essence there is not enough new information in figure 1 and 2 to justify 2 figures.
- It is already known that Glucose uptake through glut1 affects mTORC1 and protein synthesis

although this is ignored by the authors.

- Downstream of Bmp2 the main transcription factors one think of are members of the Smad family, why were they not studied here? this seems to be a missed opportunity. The study performed in Figure 7 is done without control of specificity.

Reviewer #3:

Remarks to the Author:

In this manuscript, Lee and colleagues use mouse genetics to examine glucose transport requirements during early endochondral bone development. The authors document that Glut1 is the most highly expressed glucose transporter in chondrocytes and that genetic ablation of Glut1 in limb mesenchyme or osteo-chondral progenitors inhibits longitudinal bone growth and chondrocyte hypertrophy. The study provides a thorough documentation of the developmental defects associated with the loss Glut1 expression and well as the regulation of Glut1 by Bmp signaling. I have only a few minor comments

- 1) The defect in chondrocyte hypertrophy, what appears to be a delay in the formation of the primary ossification center, and the indication that Bmp regulates Glut1 via Hif1 begs the question of whether vascular invasion of the anlage is impaired with in the Glut 1 mutants.
- 2) The whole mount used for E16.5 mutants does not appear to well represent the phenotype evident in the mutants.
- 3) There are a few instances in the Results section where references appear to be missing. The authors state "Previous studies have shown that...", but no reference is included.

We thank the reviewers for the positive comments as well as the constructive criticism. We have substantially revised the paper accordingly. Main revisions in the text are marked in red. Below we provide point-to-point reply to the critique.

Reviewer #1 (Remarks to the Author):

Minor point

Figure 7: Quantification of Western blot data should be provided.

We have now provided quantification to all Western blot data.

Reviewer #2 (Remarks to the Author):

- The expression of Glut1 in chondrocytes starting at E10.5, i.e., much earlier than in this paper, and with many more molecular markers has already been published (Wei et al., Figure 1) it is disconcerting that this is not acknowledged by the authors when they describe their own expression study/ Is there a reason for this omission?

The reviewer is correct that Wei et al documented the expression of Glut1 mRNA by in situ hybridization at multiple time points of limb development. We regret the previous oversight but have now acknowledged the work in the context of our expression study (Page 7). Different from the previous work, our study examined Glut1 protein expression by immunostaining. We began with E12.5 as this is when discrete cartilage elements can be reliably discerned in the embryonic limb. While Wei et al showed that at E10.5 (before chondrocytes form) Glut1 mRNA was diffusely detectable in the limb bud mesenchyme, it was greatly upregulated in the limb cartilage at E12.5. This prominent expression of the mRNA in chondrocytes is consistent with what we saw with the protein.

- If this study has to be repeated it should be repeated at earlier stage of cell differentiation when Prx-Cre will be active, i.e., before E13.5. It should include molecular makers of chondrocyte differentiation not only structural genes, e.g., Sox9, Sox5, Sox6.

We have repeated the study with E12.5 embryos and now include the new data in the revision. The results show that no morphological (Fig. 2P-R) or molecular defect (Sox9) (Fig. S2A) was obvious in the mutant even though Glut1 deletion was efficient at this stage. Proliferation of the Sox9+ cells was also not affected in the mutant (Fig. S2B, C). The additional data therefore support our conclusion that Glut1 is most important for chondrocyte proliferation and hypertrophy at a later stage.

- A conceptual flaw of this study is that the analysis performed by the authors starts at E14.5 in Figure 2 and contains only one panel at E12.5 in Figure 1. This is a problem for two reasons, the first one is that that stage and in agreement with the pattern of

expression of Prx itself there is already phenotypic abnormalities and so the reader is left with an unanswered question: Until when development happens normally? The second reason is that Prx-Cre is not really active beyond E14.5 and all we looking at are consequences of events occurring earlier.

We have now analyzed E12.5 embryos and show that cartilage development is normal up to that stage (Fig. 2P-R, Fig. S2A-C). The results indicate that Glut1 is dispensable for the initial chondrogenesis, likely due to compensation by the other Glut transporters. Those other transporters however are clearly inadequate to support the subsequent growth. The fact that Glut1 deletion did not affect early chondrogenesis indicates that the late effect cannot be explained by a general requirement of Glut1 by all cells, but rather is due to specific needs by chondrocytes at a later stage. This conclusion is further confirmed by the similar phenotype caused by Col2-Cre, which deletes Glut1 only after chondrogenesis has occurred.

- The authors state strongly that osteoblast differentiation is not affected in their model and yet a few lines later they acknowledge, rightly so, that Prx-cre may act indirectly and that there are limitations with this mouse. So much so that they are forced to use other Cre drivers. In fact osteoblast differentiation when it is studied, is affected in this mouse model since Type I collagen expression is abolished but this point is ignored for reasons that are unclear.

In addition to Prx1-Cre, we only used Col2-Cre to delete Glut1 in the current study. In both models, a bone collar was evident at E16.5 by histology despite abnormal morphology. We have only performed further analyses of osteoblasts with the Prx1-CKO mutant (Fig. 5). Picrosirius red staining showed that Type I collagen was readily detectable in the mutant (Fig. 5C).

- In essence there is not enough new information in figure 1 and 2 to justify 2 figures.

We respectfully submit that Figure 1 not only details Glut1 protein expression in the developing cartilage for the first time to our knowledge, but also documents the gene deletion efficiency at the different stages. Figure 2 provides a comprehensive survey of skeletal phenotypes at different stages.

- It is already known that Glucose uptake through glut1 affects mTORC1 and protein synthesis although this is ignored by the authors.

We are well aware of the elegant study by Wei et al that delineates a mechanism downstream of glucose uptake in osteoblast lineage cells. However, the present study deals with the upstream events leading to Glut1 induction in chondrocytes. The data indicate that mTORC1 is a critical mediator for Bmp to induce Glut1 expression.

- Downstream of Bmp2 the main transcription factors one think of are members of the Smad family, why were they not studied here? this seems to be a missed opportunity. The study performed in Figure 7 is done without control of specificity.

We thank the reviewer for this suggestion. We have now examined the involvement of Smad signaling by knocking down Smad4. The data show that Smad4 knockdown diminished Glut1 induction by Bmp2 (Fig. 8D). To assess whether Smad signaling might directly stimulate Glut1 transcription, we searched the Smad4 ChIP-seq database published by Yan et al (J. Biol. Chem. (2018) 293(24) 9162–9175). The authors identified 1213 genes with Smad4 binding peaks within 5 kb of the transcription start site in E12.5 and E13.5 limb cartilage, but did not observe such binding on the Glut1 locus, indicating that Glut1 may not be a direct target gene of Smad4. On the other hand, we found that Smad4 knockdown diminished mTORC1 activation by Bmp2, indicating that the effect of Smad signaling may be mediated by mTORC1 (Fig. 8F). As Smad4 has been reported to suppress Pten, an upstream negative regulator of mTORC1 in other systems (Xu et al., 2006, J. Clin. Invest. 116:1843–1852; Xiong et al., Oncotarget 7: 61262-61272), we proceeded to examine the Pten mRNA level in chondrocytes in response to Bmp2 with or without Smad4 knockdown. We found that Bmp2 consistently suppressed Pten mRNA by ~20% in a Smad4-dependent manner, indicating that Pten suppression may partially explain the activation of mTORC1 by Smad signaling. Therefore, collectively the data supports a model wherein Bmp activates mTORC1 through Smad4 and perhaps other mediators to enhance Hif1a translation and subsequently Glut1 transcription.

--

Reviewer #3:

1) The defect in chondrocyte hypertrophy, what appears to be a delay in the formation of the primary ossification center, and the indication that Bmp regulates Glut1 via Hif1 begs the question of whether vascular invasion of the anlage is impaired with in the Glut 1 mutants.

We have now provided endomucin immunostaining to show that vascular invasion tracks the delay in forming primary ossification center (Fig. 5B).

2) The whole mount used for E16.5 mutants does not appear to well represent the phenotype evident in the mutants.

Whole mount staining of E16.5 mutant embryos detected little to no mineral staining in the scapula or the humerus, whereas the other skeletal elements exhibited a decrease in staining. We observed the same pattern in 3 out of 3 mutants compared with their littermate controls. The images in Fig. 2F-J are representative of all E16.5 embryos analyzed by whole mount staining. Please note that all subsequent studies of sections were performed with the femur.

3) There are a few instances in the Results section where references appear to be missing. The authors state “Previous studies have shown that...”, but no reference is included.

We have corrected this (page 11).

Reviewers' Comments:

Reviewer #2:

Remarks to the Author:

The authors have adequately answered my queries, I have no more concerns.